# Emergency department utilization for substance use disorders and mental health conditions during COVID-19

Arjun K. Venkatesh[1,2]⊙*, Alexander T. Janke[1]⊙*, Jeremy Kinsman[1]‡, Craig Rothenberg[1]‡, Pawan Goyal[3]‡, Caitlin Malicki[1]‡, Gail D'Onofrio[1]‡, Andrew Taylor[1]‡, Kathryn Hawk[1,4]‡

**1** Department of Emergency Medicine, Yale University School of Medicine, New Haven, Connecticut, United States of America, **2** Center for Outcomes Research and Evaluation, Yale University School of Medicine, New Haven, Connecticut, United States of America, **3** American College of Emergency Physicians, Irving, Texas, United States of America, **4** Yale School of Public Health, New Haven, Connecticut, United States of America

⊙ These authors contributed equally to this work.
‡ These authors also contributed equally to this work.
* arjun.venkatesh@yale.edu (AKV); alexander.janke@yale.edu (ATJ)

**Data Availability Statement:** The data underlying the results presented in the study are from the Clinical Emergency Data Registry from the American College of Emergency Physicians

## Abstract

### Background

As the emergency department (ED) has evolved into the de-facto site of care for a variety of substance use disorder (SUD) presentations, trends in ED utilization are an essential public health surveillance tool. Changes in ED visit patterns during the COVID-19 pandemic may reflect changes in access to outpatient treatment, changes in SUD incidence, or the unintended effects of public policy to mitigate COVID-19. We use a national emergency medicine registry to describe and characterize trends in ED visitation for SUDs since 2019.

### Methods

We included all ED visits identified in a national emergency medicine clinical quality registry, which included 174 sites across 33 states with data from January 2019 through June 2021. We defined SUD using ED visit diagnosis codes including: opioid overdose and opioid use disorder (OUD), alcohol use disorders (AUD), and other SUD. To characterize changes in ED utilization, we plotted the 3-week moving average ratio of visit counts in 2020 and 2021 as compared to visit counts in 2019.

### Findings

While overall ED visits declined in the early pandemic period and had not returned to 2019 baseline by June 2021, ED visit counts for SUD demonstrated smaller declines in March and April of 2020, so that the proportion of overall ED visits that were for SUD increased. Furthermore, in the second half of 2020, ED visits for SUD returned to baseline, and increased above baseline for OUD ever since May 2020.

(ACEP). ACEP can be contacted via the following link to request information related to confidential data: https://webapps.acep.org/membership/account/#/messageus?c=true&url=https:%2F%2Fwww.acep.org%2Fcedr%2F&app=CEDR&to=U2FsdGVkX18ws8mNv3WgjkppwH8koQH32hvHJ7pQsdk%3D.

**Funding:** This work was supported by the HHS Office of the Secretary Patient Centered Outcomes Research Trust Fund (PCORTF) under IDDA# ASPE-2018-001 and NIDA UG1DA015831-18S2. In addition, AKV was supported by KL2 TR000140 from the National Center for Advancing Translational Sciences of the NIH and the American Board of Emergency Medicine – National Academy of Medicine Anniversary Fellowship. The contents of this work are solely the responsibility of the authors and do not necessarily represent the official view of NIH. The funders had no role in study design, data collection and analysis, decision to publish, or preparation of the manuscript.

**Competing interests:** The authors have declared that no competing interests exist.

## Conclusions

We observe distinct patterns in ED visitation for SUDs over the course of the COVID-19 pandemic, particularly for OUD for which ED visitation barely declined and now exceeds previous baselines. These trends likely demonstrate the essential role of hospital-based EDs in providing 24/7/365 care for people with SUDs and mental health conditions. Allocation of resources must be directed towards the ED as a de-facto safety net for populations in crisis.

## Introduction

The Centers for Disease Control and Prevention (CDC) first documented a 42% decline in emergency department (ED) visits across the United States (US) during the early COVID-19 pandemic [1]. Decreased ED presentations for medical emergencies such as acute myocardial infarction (MI) and stroke, for which care-seeking may be discretionary in those with occult or atypical symptoms, have also been documented [2]. However, emergency medical services (EMS) and syndromic surveillance data has suggested an increase in opioid overdoses during the early months of the pandemic. In some regions, this may reflect worsening access to substance use disorder (SUD) treatment options amidst state and local shutdowns or the psychosocial effects of COVID-19 that place people at higher risk of overdose or returning to use [3–7]. Prior to the identification of COVID-19, the US was facing the highest reported number of drug overdose deaths ever reported, with 72,224 drug overdose deaths provisionally reported by the CDC in the twelve months preceding January 2020 [8]. More recently, the CDC has provisionally released that 96,779 drug overdose deaths have occurred in the twelve months preceding the end of March 2021, a new record for the highest number of drug overdose deaths in a twelve-month period ever recorded. Despite this, work characterizing changes in ED visits for SUD have documented decreased visit rates through July in a large contract management group sample of 108 EDs across 18 states, with a more muted decline in arrivals by EMS for opioid related reasons [9]. Other work using administrative claims data found similar declines in ED visits for substance use disorders through April 20, 2020 [10], and work using the National Syndromic Surveillance Program data found that ED visits were above the 2019 baseline for mental health, suicide attempts, overdose, and violence through October, 2020 [11]. While these concurrent crises have been described as an epidemic within a pandemic, little is known about how ED presentations for a broader array of SUDs or for mental health conditions, known to cooccur in nearly 40% of Americans with SUD [12,13], have been impacted during the evolving pandemic into 2021.

Given the gradual emergence of a 'new normal' and ongoing concerns about excess mortality both directly attributable to COVID-19 and related to SUDs, we aimed to provide an updated characterization of ED visitation for SUD in a national sample of hospital-based EDs and differentiate from prior work with a more granular exploration of visits for OUD as well as mental health conditions in comparison to commonly studied emergent medical conditions. As the ED has evolved into the de-facto site of care for a variety of SUD presentations including overdose, intoxication, acute withdrawal, and related mental health conditions, and given the finding that SUD patients were much more likely to present to the ED with COVID-19 [14], trends in ED visits are an essential public health surveillance tool for monitoring the prevalence and severity of SUD and facilitating ED-based efforts at treatment initiation and linkage [15]. Changes in ED visit rates for SUD may reflect changes in access to outpatient treatment for SUD or mental health conditions, broader trends in SUD incidence, the

intended or unintended impact of COVID-19 public policy, or the effect of initiatives to address SUD through better treatment and social service coordination.

Accordingly, we utilized a national emergency medicine clinical quality registry of aggregated electronic health record data to describe trends in ED visitation for SUDs since 2019 through the current COVID-19 pandemic across a wide range of clinical and geographic settings.

## Methods

### Study design

We conducted an observational study of ED visits included in national emergency medicine clinical quality registry with data from January 1, 2019 through June 30, 2021.

### Study setting and dataset

We included all ED visits identified in the Clinical Emergency Data Registry (CEDR), a voluntary national emergency medicine clinical quality registry maintained by the American College of Emergency Physicians (ACEP) with complete data from January 1, 2019 through June 30, 2021. In general, the EDs participating in the ACEP CEDR reflect community, hospital-based EDs of similar regional, case and insurance mix to national datasets with notably fewer academic hospitals. In order to characterize hospital-based EDs in the sample according to their urban-rural status, as well as their status as an academic hospital (an Accreditation Council for Graduate Medical Education accredited program), control status (public, private for-profit, not-for-profit), number of inpatient beds, sites were linked to data from the 2018 American Hospital Association Annual Survey, the most recently available data [16]. We furthermore linked this registry data to publicly available data on daily active COVID-19 cases per 1,000 population from the University of Maryland COVID-19 Impact Analysis Platform [17] to estimate the burden of COVID-19 in the county where each hospital-based ED was located.

### Definitions

We defined SUD conditions based on ED diagnosis of *International Classification of Disease*, *Tenth Revision*, *Clinical Modification* (ICD-10-CM) diagnosis codes. Consistent with prior work, we defined SUD using diagnosis code groupings of the *Clinical Classification Software Revised* (CCSR) system, maintained by the Healthcare Cost and Utilization Project [9,18] to provide useful groupings of diagnosis codes, as well as select subsets of public health interest within that group including opioid overdose and opioid use disorder (OUD) (MBD018) and alcohol use disorders (AUD) (MBD017) [19], as well as other SUD inclusive of cannabis (MBD019), sedative/hypnotic/anxiolytic (MBD020), cocaine/other stimulant (MBD021), hallucinogenic (MBD022), inhalant (MBD023), and other psychoactive substance-related disorders (MBD025). As reference and to characterize changes in ED visitation for SUDs, we also characterized ED visit counts for two common acute medical emergencies, acute myocardial infarction and stroke (CCSR codes *CIR009* and *CIR020*). Given the common co-occurrence of mental health conditions alongside SUDs, and evidence that conditions such as depression and anxiety that often cooccur with SUD and have escalated in incidence during COVID-19 due to social isolation, unemployment and other factors [20], we also identified ED visits for mental health conditions, broadly defined. We identified mental health conditions according to *International Classification of Disease*–10 (ICD-10) root codes F2 through F9, inclusive of psychotic and mood disorders, behavioral syndromes, and disorders of personality. **S1 Table**.

## Outcomes

The primary outcome was the weekly counts of ED visits for SUDs including subsets of OUD and AUD. As secondary outcomes to provide additional context we also calculated weekly counts of ED visits for acute myocardial infarction, stroke, and mental health conditions.

## Analysis

To characterize changes in ED utilization for each condition, we plotted the 3-week moving average ratio, and associated 95% confidence intervals, of ED visit counts in 2020 and 2021 as compared to the same weeks in 2019. Due to the dramatic changes in overall ED visit counts throughout the pandemic, we identified different periods for descriptive analysis by applying a segmented linear regression approach on overall ED visit counts, with time as the only input into the model, using a previously described iterative linearization process for finding unknown breakpoints (**S2 Fig**) [21]. These simply-derived breakpoints were used to identify time periods for subsequent descriptive analysis. Given distinct epidemiologic implications, we also report ED utilization in each of these identified periods for each condition in three ways including: 1) average weekly counts, 2) proportion of overall ED visits, 3) hospital-based ED visit incident rate ratios (IRR) for weekly visit counts comparing different periods in 2020/ 2021 to identical periods in 2019. IRRs for site-level ED visit counts across conditions, with 95% confidence intervals, were obtained via unadjusted Poisson regression models for hospital-based daily ED visit counts in each period compared to the same weeks in 2019.

As a secondary analysis to examine the impact of known geographic and temporal variation in COVID-19 outbreaks across the US, which may impact care-seeking for SUD, we conducted additional analyses stratified by geographic regions, defined according to census regions [22]: Northeast, Midwest, South, West. We utilized the daily active COVID-19 cases per 1,000 population for the county of each ED, averaged across counties represented in our sample in each census region, and plotted these along with SUD to provide context for trends in ED visits. Analyses were performed in R (4.0.2). Access to the CEDR data is restricted by agreement with ACEP, given potentially identifying features, and the study was classified as exempt by the Institutional Review Board at Yale University.

## Results

Among a total of 174 EDs across 33 states included (**S1 Fig**), median ED visit volumes in 2019 was 29,296 [interquartile range 15,989 to 47,082], with the most sites in California (41) and Texas (15). Of all EDs, 55 (31.6%) are located in rural areas while 119 (68.4%) are located in urban areas [23,24]. 79 (45.4%) were sites participating in an Accreditation Council for Graduate Medical Education accredited program and 95 (54.6%) did not. 83 (47.7%) were nongovernment, not-for-profit, 33 (19.0%) were for-profit entities, 21 (12.1%) were church-operated, and the remaining 37 (21.2%) were various nonfederal government entities. Median inpatient beds was 138, with a minimum of 11 and maximum of 1,031 (interquartile range 49 to 239). Across sites, the proportion of visits covered by Medicare, Medicaid, and private insurance were 25.7%, 26.9%, and 24.6%, respectively.

As in **Fig 1**, overall ED visit counts declined in the early pandemic period of March and April 2020, to a nadir of 52% of levels in the same period in 2019, before returning to 75–85% of visit counts in the period from July 2020 through June 2021. ED visits for SUD and mental health exhibited declines in counts as well, to 72% and 71%, respectively, much closer to 2019 levels than overall visits throughout the second half of 2020 and into 2021. Shown in **Fig 2**, while visit counts for AUD remained low relative to 2019, ED visits for OUD returned to 2019 levels by May, and were subsequently above 2019 counts for much of 2020 and into 2021.

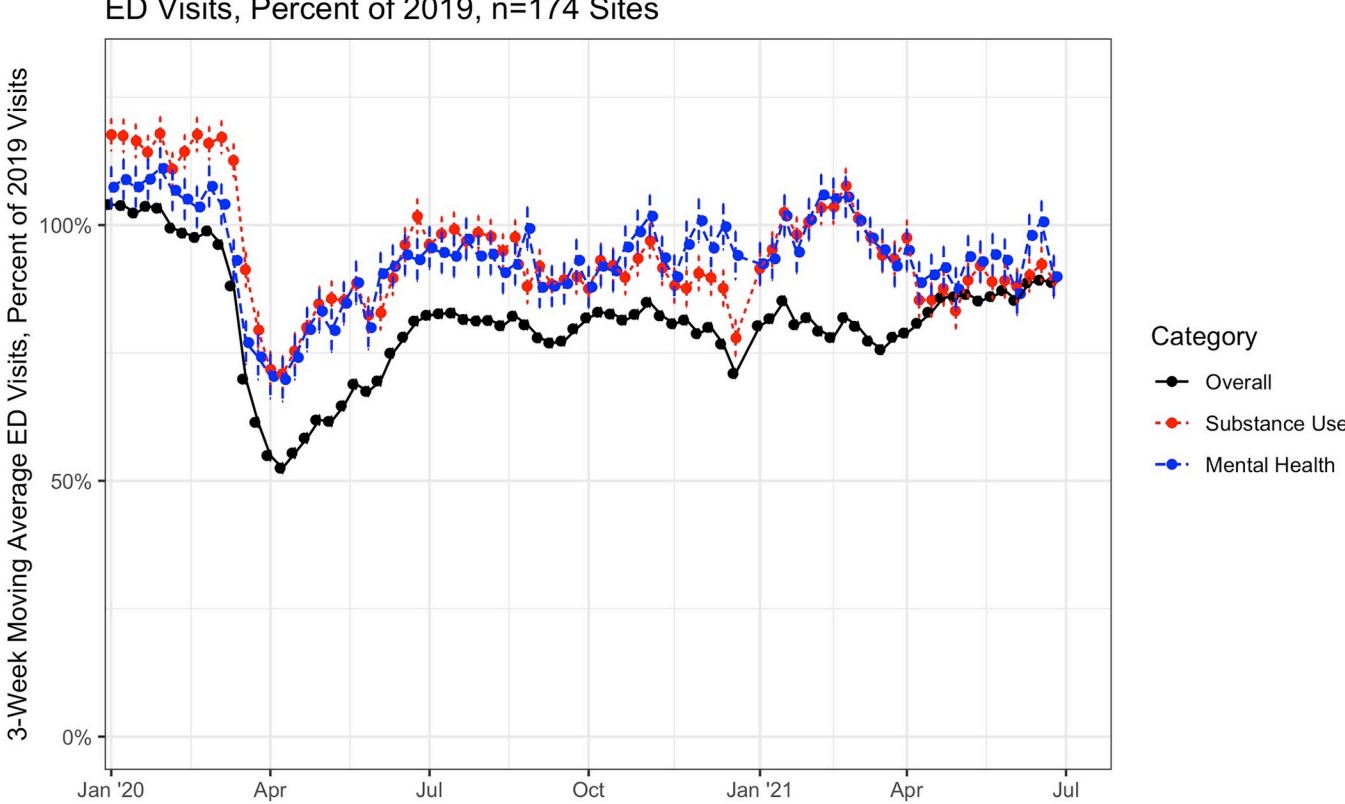

**Fig 1. Trends in weekly ED visits for substance use disorders and mental health conditions, 2020.** Notes: Figure depicts 3-week moving average ratio of emergency department visit counts in 2020 and 2021 compared to identical weeks in 2019. Sample includes 174 hospital-based EDs across 33 states from a national emergency department clinical quality registry. Visits for substance use disorders and mental health conditions were identified by *International Classification of Disease*-10 codes.

Table 1 depicts weekly average visit counts, the proportion of overall ED visits for each diagnosis group, and the incident rate ratio for those visits as compared to the same weeks in 2019. Weekly ED visit counts for OUD and other SUD (non-alcohol and non-opioid) were statistically significantly above the 2019 baseline counts in the first 12 weeks of 2021 (IRR 1.06 [1.05–1.08] and IRR 1.03 [1.01–1.04], respectively), while visits for alcohol remained lower (IRR 0.94 [0.92–0.95]). As with OUD, ED visit counts for mental health, as well as acute myocardial infarction and stroke, were at or above the 2019 levels in the final periods of 2020. Visit counts were lower following the arrival of COVID-19 for each of OUD, AUD, and other SUD (S3 Fig), but visits rose as a proportion of overall ED visits (S4 Fig).

The secondary analysis of ED visitation based on geography demonstrated similar patterns for both overall ED visits as well as for overall SUD, AUD, and OUD across each regions despite substantially different covid outbreak patterns. In general, visit counts for SUD were high relative to 2019 in January and February, 2020, especially in the Northeast and Midwest, and fell to below 2019 levels briefly before again rising, in the second half of 2020. **Fig 3.**

## Discussion

Unlike overall ED visitation, which has been shown to have substantially declined and then returned to modestly lower counts during the COVID-19 pandemic relative to 2019, declines in ED utilization for SUD were more muted and rebounded to 2019 levels earlier than even

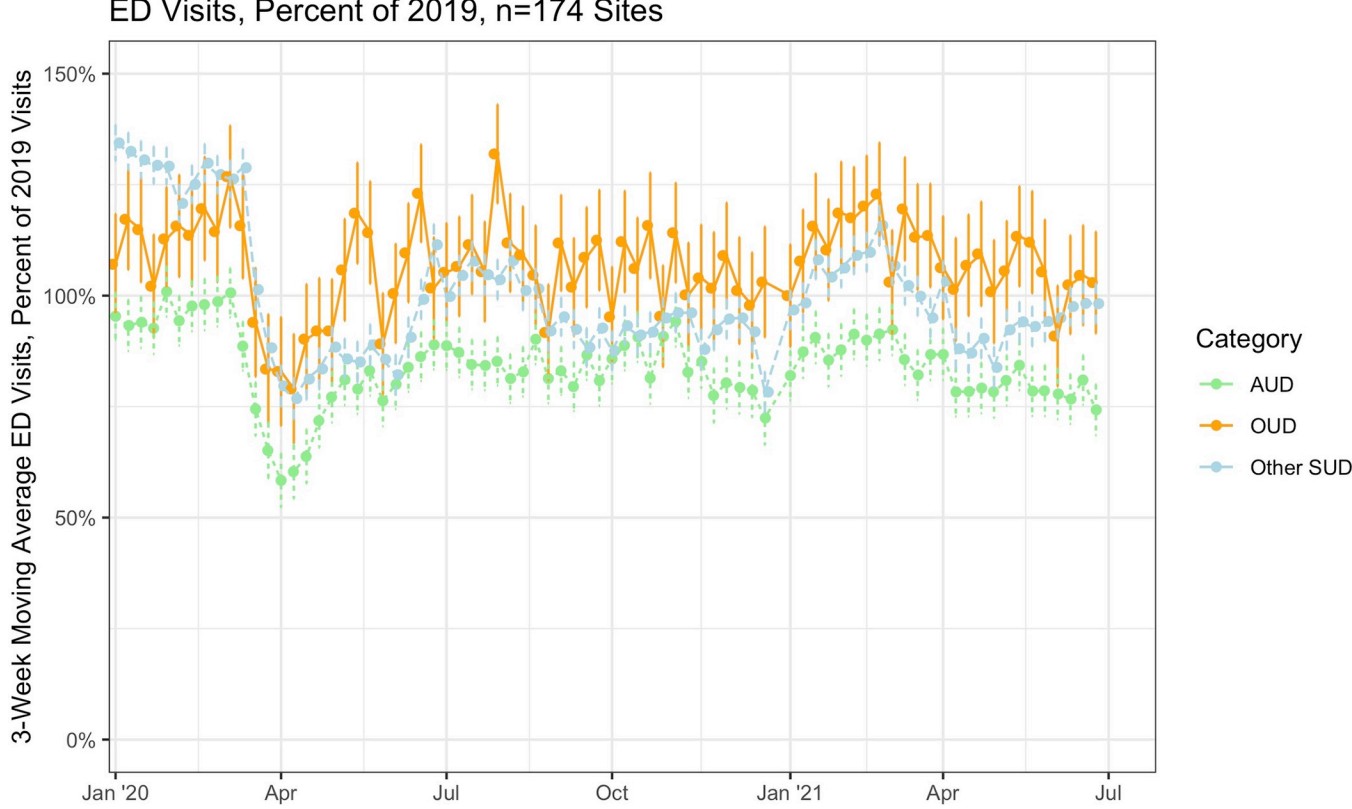

**Fig 2. Trends in weekly ED visits for OUD, AUD, and other SUD-related, 2020.** Notes: Figure depicts 3-week moving average ratio of emergency department visit counts in 2020 and 2021 compared to identical weeks in 2019. Sample includes 174 hospital-based EDs across 33 states from a national emergency department clinical quality registry. Visits for opioid use disorder (OUD), alcohol use disorder (AUD) and other substance use disorder (SUD)-related were identified by *International Classification of Disease*-10 codes.

emergency conditions, such as acute MI and stroke [2,25]. Furthermore, despite the common cooccurrence of SUDs, ED visits for OUD have been increasing at a substantially higher rate than AUD or other SUDs since the beginning of the COVID-19 pandemic. This suggests that the pandemic may have disproportionate condition specific impacts as a result of fundamentally different patterns of acute care access and outcomes for different clinical scenarios. For example, while the pandemic may have resulted in impeded access to outpatient treatment programs (OTPs) or office-based buprenorphine prescriptions that in turn resulted in poor OUD management and increased ED visitation, the pandemic may not have had as much relative effect on services for AUD that are less medication or healthcare facility dependent. It is likely that a shift of focus away from the opioid epidemic as systems sought to manage and mitigate the COVID-19 pandemic may have also expanded supply chain gaps, worsened contacts with healthcare providers necessary to ensure access to naloxone, potentially driving increases in opioid-associated fatalities and exacerbating inequities in access to care.

Interestingly, we found little variation in ED utilization trends for each condition based on geography despite substantial variation in covid outbreak timing, public health policies and known geographic variability in SUD incidence. This likely supports the notion that initial declines in ED visitation reflect a broad change in care seeking behavior as opposed to a clinically nuanced decision by people. This finding also demonstrates that COVID burdens of EDs and hospitals were not related to local care seeking patterns or the local epidemiology of SUD. CDC provisional death data, though, show geographic variability in the burden of SUD

**Table 1. ED utilization for substance use disorder and mental health.**

| | | 2020 | | | | 2021 | |
|---|---|---|---|---|---|---|---|
| | | Weeks 1–10 | Week 11–13 | Weeks 14–27 | Weeks 28–53 | Weeks 1–12 | Weeks 13–25 |
| **SUD** | Weekly Average Counts | 7977 | 6651 | 6162 | 6581 | 6865 | 6423 |
| | Proportion | 6.81% | 7.64% | 8.06% | 7.20% | 7.35% | 6.50% |
| | IRR | 1.16 | 0.95 | 0.85 | 0.92 | 0.99 | 0.94 |
| | IRR 95% CI | 1.12–1.2 | 0.79–1.14 | 0.81–0.89 | 0.89–0.95 | 0.98–1.01 | 0.92–0.96 |
| **Alcohol** | Weekly Average Counts | 2360 | 1964 | 2097 | 2194 | 2183 | 2142 |
| | Proportion | 2.02% | 2.26% | 2.74% | 2.40% | 2.34% | 2.17% |
| | IRR | 0.97 | 0.76 | 0.77 | 0.84 | 0.94 | 0.89 |
| | IRR 95% CI | 0.92–1.01 | 0.63–0.93 | 0.72–0.82 | 0.81–0.87 | 0.92–0.95 | 0.88–0.91 |
| **Opioid** | Weekly Average Counts | 619 | 517 | 579 | 607 | 610 | 595 |
| | Proportion | 0.53% | 0.59% | 0.76% | 0.66% | 0.65% | 0.60% |
| | IRR | 1.14 | 0.97 | 1.00 | 1.06 | 1.06 | 1.01 |
| | IRR 95% CI | 1.10–1.18 | 0.86–1.10 | 0.93–1.08 | 1.01–1.12 | 1.05–1.08 | 0.99–1.04 |
| **Other SUD-related** | Weekly Average Counts | 4998 | 4169 | 3486 | 3780 | 4072 | 3687 |
| | Proportion | 4.27% | 4.79% | 4.56% | 4.14% | 4.36% | 3.73% |
| | IRR | 1.29 | 1.06 | 0.88 | 0.95 | 1.03 | 0.96 |
| | IRR 95% CI | 1.25–1.33 | 0.88–1.29 | 0.84–0.92 | 0.92–0.98 | 1.01–1.04 | 0.94–0.98 |
| **Mental Health-related** | Weekly Average Counts | 5050 | 3981 | 4037 | 4350 | 4700 | 4427 |
| | Proportion | 4.31% | 4.58% | 5.28% | 4.76% | 5.03% | 4.48% |
| | IRR | 1.07 | 0.82 | 0.84 | 0.94 | 1.02 | 0.96 |
| | IRR 95% CI | 1.05–1.09 | 0.71–0.94 | 0.80–0.87 | 0.90–0.98 | 0.98–1.01 | 0.94–0.99 |
| **MI and Stroke** | Weekly Average Counts | 1166 | 896 | 966 | 1081 | 1160 | 1063 |
| | Proportion | 1.00% | 1.03% | 1.26% | 1.18% | 1.24% | 1.08% |
| | IRR | 0.97 | 0.75 | 0.86 | 0.97 | 0.98 | 0.97 |
| | IRR 95% CI | 0.94–1.00 | 0.70–0.80 | 0.83–0.90 | 0.95–0.99 | 0.97–1.00 | 0.95–0.99 |
| **Overall** | **Weekly Average Visits** | 117,064 | 87,011 | 76,457 | 91,414 | 93,446 | 98,776 |

Note: Sample includes 174 hospital-based EDs across 33 states from a national emergency department clinical quality registry. Visits for different diagnosis categories were identified by *Clinical Classification Software Revised* and *International Classification of Disease*-10 codes. Incident rate ratios (IRRs) are estimated with unadjusted Poisson regression models for hospital-based ED daily visit counts in each period as compared to the same weeks in 2019. SUD = substance use disorder, MI = myocardial infarction.

mortality, especially with disproportionate impact in California, Arizona, and Washington and much policy attention to addressing the crisis [26]. Interestingly, amongst the four US census regions reported in **Fig 3**, the West exhibited the smallest increases in ED presentations for OUD in our analysis. While this may represent a limitation of our work and ED visitation in our sample may not be representative generally, California was the best represented state in our sample (with 41 hospital-based EDs). This disconnect between ED visitation and population-level mortality may be related to delayed activation of EMS services. Given the value of EDs as sites for the initiation of medication-assisted therapy for OUD [27], failure to initiate transport in cases of opioid or other overdose not only risks mortality in the acute phase but also risks failure to initiate treatment and avoid a subsequent overdose.

Several phenomena may explain these findings. First, given the importance of the ED in providing access to care for SUDs in both the acute phase, such as overdose or skin abscess, as well as initiating treatment for SUDs and serving as a conduit to limited outpatient SUD treatment resources, people with SUDs likely continued to utilize the ED to access services. This was likely compounded during the COVID-19 pandemic by the commensurate closure or

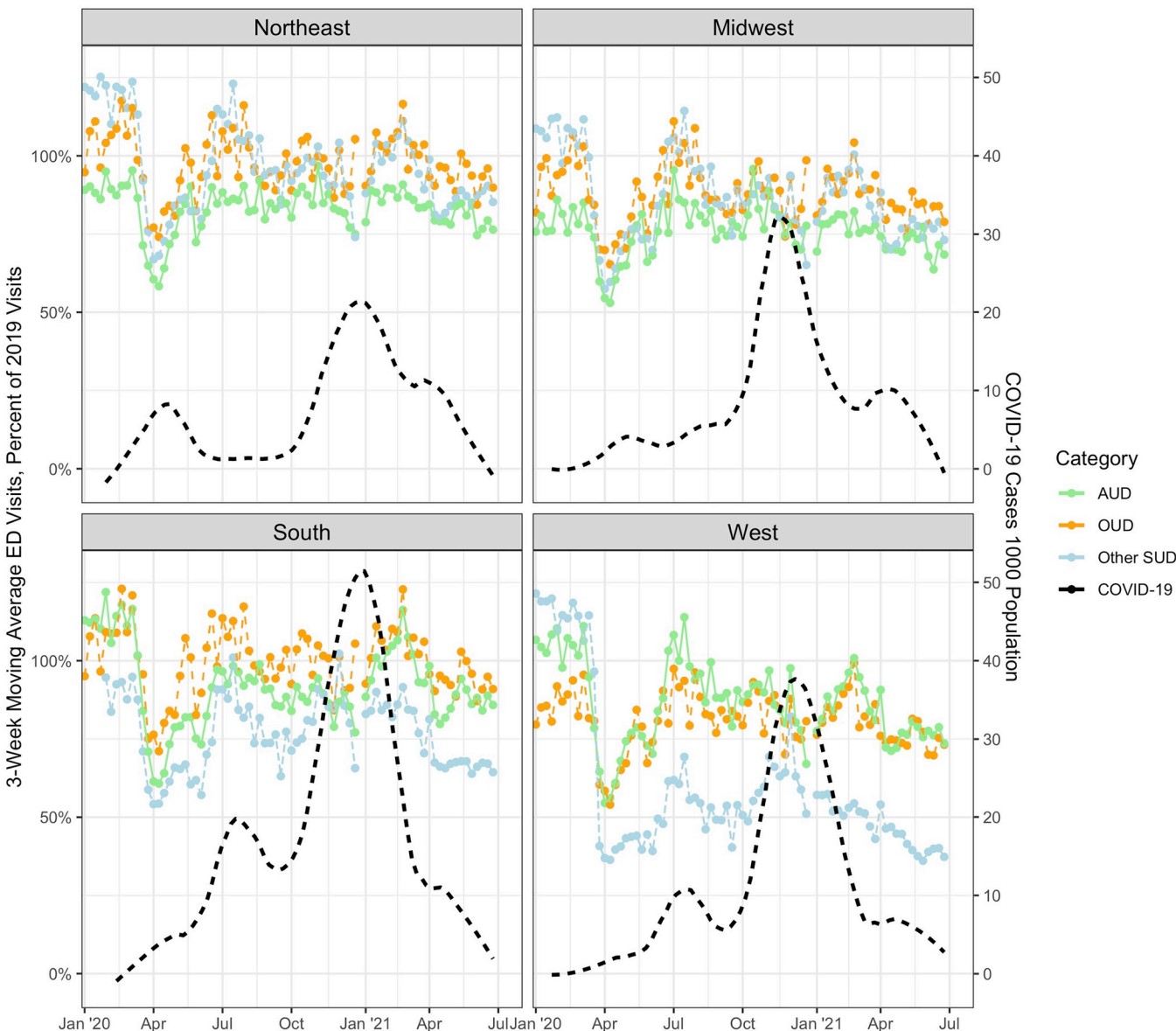

**Fig 3. Trends in weekly ED visits for OUD, AUD, and other SUD-related, 2020, By US census regions.** Note: Figure depicts 3-week moving average ratio of emergency department visit counts in 2020 and 2021 compared to identical weeks in 2019 in each U.S. Census Region. Sample includes 174 hospital-based Eds across 33 states from a national emergency department clinical quality registry. Visits for opioid use disorder (OUD), alcohol use disorder (AUD) and other substance use disorder (SUD)-related were identified by International Classification of Disease-10 codes.

limitation of many outpatient care venues because of government shutdown and social distancing policies. Second, the relatively higher utilization of the ED for SUDs, particularly OUD, may reflect compensatory increases in substance use or exacerbations of mental health conditions because of social policies. Furthermore, patients with injection drug use may have had increasingly limited access to harm reduction services and sterile syringes, leading to increased skin and soft tissue infections for which ED care was needed. In fact, our analysis may underestimate the degree of increased SUD associated harms as EMS data during the pandemic suggest both increased transport refusals for a myriad of conditions as well as increased out-of-hospital overdose-related cardiac arrests, which may not be classified as opioid-related

and may have resulted in transport to the ED previous to the pandemic [7,28]. Third, our findings may represent less patient discretion in seeking emergency care for acute symptoms and concerns related to SUDs than even other medical emergencies. For example, a person experiencing an overdose may be more likely to be involuntarily transported to an ED in comparison to a person with chest pain who may underestimate the clinical significance of symptoms indicative of an acute myocardial infarction and overestimate the infectious risk of seeking care in hospital EDs.

Importantly, increasing social isolation and unemployment associated with COVID-19 may be exacerbating US rates of "deaths of despair," which has been identified as the etiology of decreased US life expectancy from 2014–2017 when midlife mortality increased across all racial groups, caused by drug overdoses, alcohol use, suicides, and a diverse list of organ system diseases [29–32], with opioid overdoses exhibiting racial and ethnic disparities as well [33]. Health systems and overburdened EDs continue to adapt operations to the COVID-19 pandemic, particularly given increased trends of extremely prolonged ED length of stay, particularly the "boarding" of ED patients in the ED awaiting inpatient bed assignment [34]. Patients may increasingly seek care for conditions previously delayed or deferred, and the essential role of the hospital-based ED in providing 24/7/365 access to care for people with SUDs and mental health conditions becomes more evident.

## Limitations

These findings should be interpreted within the confines of the study dataset and design. First, our sample includes primarily community based EDs similar to most EDs and hospitals in the US, however may not be generalizable to teaching or specialty centers that are disproportionately less likely to participate in the ACEP CEDR registry. Second, while we acknowledge the likely important interaction between race and ethnicity and these observed trends, the ACEP CEDR registry does not currently capture this data reliably and future work should explore whether observed changes in visitation trends reflect widening disparities in access to SUD care for disadvantaged populations. Third, our dataset is limited to examining ED visitation without an ability to capture other concurrent care setting such as telemedicine that have increased access to care for select SUD populations during the pandemic [35]. However, given that among SUDs OUD is likely the most amenable to telemedicine management (e.g ability to initiate and prescribe buprenorphine), but we observed the greatest increase in ED visits for OUD, these innovative care models are not likely to have had widespread impact.

## Conclusions

As health systems continue to adapt to the COVID-19 pandemic, including substantial changes in ED and hospital volumes for SUD care which may have been delayed, deferred or exacerbated by the pandemic, the essential role of the hospital-based ED in providing 24/7/365 care for people with SUDs and mental health conditions has heightened importance. Allocation of resources must be directed towards the ED as a de-facto safety net that minimizes the morbidity and mortality for populations in crisis.

## Supporting information

**S1 Fig. ACEP clinical emergency data registry sample sites.**
(TIF)

**S2 Fig. Trends in overall weekly ED visits with segmented linear regression.**
(TIF)

**S3 Fig. Visit counts for SUD: OUD, AUD, and other SUD.**
(TIF)

**S4 Fig. Proportion of overall visits for SUD: OUD, AUD, and other SUD.**
(TIF)

**S1 Table.**
(DOCX)

## Author Contributions

**Conceptualization:** Arjun K. Venkatesh, Alexander T. Janke.

**Data curation:** Alexander T. Janke.

**Formal analysis:** Alexander T. Janke, Craig Rothenberg.

**Funding acquisition:** Arjun K. Venkatesh.

**Investigation:** Arjun K. Venkatesh, Alexander T. Janke, Craig Rothenberg.

**Methodology:** Arjun K. Venkatesh, Alexander T. Janke.

**Project administration:** Jeremy Kinsman, Gail D'Onofrio.

**Resources:** Jeremy Kinsman, Caitlin Malicki, Andrew Taylor.

**Software:** Alexander T. Janke, Craig Rothenberg.

**Supervision:** Pawan Goyal, Gail D'Onofrio, Andrew Taylor, Kathryn Hawk.

**Visualization:** Alexander T. Janke.

**Writing – original draft:** Alexander T. Janke.

**Writing – review & editing:** Arjun K. Venkatesh, Alexander T. Janke, Craig Rothenberg, Pawan Goyal, Caitlin Malicki, Andrew Taylor, Kathryn Hawk.

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
