## [Decision Letter · Decision Letter 0]

20 Sep 2021

PONE-D-21-20023Emergency Department Utilization for Substance Use Disorders and Mental Health Conditions During COVID-19PLOS ONE

Dear Dr. Venkatesh,

Thank you for submitting your manuscript to PLOS ONE. After careful consideration, we feel that it has merit but does not fully meet PLOS ONE’s publication criteria as it currently stands. Therefore, we invite you to submit a revised version of the manuscript that addresses the points raised during the review process.

Please address the following journal requirements and additional editor and reviewer comments.

We look forward to receiving your revised manuscript.

Kind regards,

Fernando A. Wilson, PhD

Academic Editor

PLOS ONE

2. In ethics statement in the manuscript and in the online submission form, please provide additional information about the patient records/samples used in your retrospective study. Specifically, please ensure that you have discussed whether all data/samples were fully anonymized before you accessed them and/or whether the IRB or ethics committee waived the requirement for informed consent. If patients provided informed written consent to have data/samples from their medical records used in research, please include this information.

“This work was supported by the HHS Office of the Secretary Patient Centered Outcomes Research Trust Fund (PCORTF) under IDDA# ASPE-2018-001 and NIDA UG1DA015831-18S2. In addition, Dr. Venkatesh was supported by KL2 TR000140 from the National Center for Advancing Translational Sciences of the NIH and the American Board of Emergency Medicine – National Academy of Medicine Anniversary Fellowship. The contents of this work are solely the responsibility of the authors and do not necessarily represent the official view of NIH.”

 “This work was supported by the HHS Office of the Secretary Patient Centered Outcomes Research Trust Fund (PCORTF) under IDDA# ASPE-2018-001 and NIDA UG1DA015831-18S2. In addition, AKV was supported by KL2 TR000140 from the National Center for Advancing Translational Sciences of the NIH and the American Board of Emergency Medicine – National Academy of Medicine Anniversary Fellowship. The contents of this work are solely the responsibility of the authors and do not necessarily represent the official view of NIH.

Additional Editor Comments (if provided):

- The manuscript is not formatted according to PLOS ONE requirements, eg, include page and continuous line numbers. Refer to the following link:

https://journals.plos.org/plosone/s/submission-guidelines

- References are also not formatted following PLOS ONE requirements. Please review each reference and verify their formatting with the following instructions. Use square brackets for citations. Use NCBI journal abbreviations. For #2, please update this reference since it’s now published.

https://journals.plos.org/plosone/s/submission-guidelines#loc-references

https://www.ncbi.nlm.nih.gov/nlmcatalog/journals

- Add a dash within “co-occurrence”.

- Use either “U.S.” or “US” consistently throughout the manuscript.

- Use “1,000” instead of “1000” to be consistent with formatting of other numbers.

- Please provide more details on the regression analyses and regression tables in the Supporting Information section.

Reviewers' comments:

Reviewer's Responses to Questions

**Comments to the Author**

1. Is the manuscript technically sound, and do the data support the conclusions?

Reviewer #1: Yes

Reviewer #2: Partly

2. Has the statistical analysis been performed appropriately and rigorously? 

Reviewer #1: Yes

Reviewer #2: I Don't Know

3. Have the authors made all data underlying the findings in their manuscript fully available?

Reviewer #1: Yes

Reviewer #2: No

4. Is the manuscript presented in an intelligible fashion and written in standard English?

Reviewer #1: Yes

Reviewer #2: Yes

5. Review Comments to the Author

Reviewer #1: In this revised paper, the authors were very responsive to two previous reviewers’ comments and suggestions and addressed their concerns. The revised paper is solid in its analyses and presents interesting findings. Figures of changes overtime (compared to the 2019 baselines) are helpful for readers. However, writing in some parts, especially in the Discussion section, is difficult to understand (as previous reviewers also pointed out) and confusing. I also have a question about the findings related to geographic differences.

1. The major source of confusion is that ED visit counts and rates are described in the same paragraph and it is not clear which one is which. I gathered that the % in the Y axis in the Figures is the proportion of ED visit counts in 2020 as compared to the same weeks in 2019. In the Results section (first paragraph—please specify page numbers), it is stated that “ED visits for SUD and mental health exhibited similar, however, more muted declines in ED visitation and were much closer to baseline than overall visits by in the second half of 2020.” And then, “While visit rates for AUD remained below 2019, ED utilization for OUD returned to 2019 levels by June, and were subsequently above 2019 baseline for the remainder of the year.” The “visit rate” is confusing, as it is not clear what you mean by that. Also clarify that the “baselines” are the ED visit counts in the corresponding weeks in 2019. Also, in “ED visits for SUD and mental health exhibited similar, however, more muted declines in ED visitation and were much closer to baseline than overall visits by in the second half of 2020,” you do not have to repeat “in ED visitation.”

2. First paragraph of the Discussion section: “…ED visits for OUD have been increasing [at is missing here] a substantial higher rate than AUD or other SUDs since the beginning of the COVID-19 pandemic. These distinct condition specific trends demonstrate the fundamentally different patterns of acute care access and outcomes for different clinical scenarios. Most importantly, it is likely that a shift of focus away from the opioid epidemic as systems sought to manage and mitigate the COVID-19 pandemic expanded gaps in OUD treatment and naloxone access, driving increases in opioid-associated fatalities and exacerbating inequities in access to care.” I found last two sentences very confusing to read, as the second sentence is not specific about which the fundamentally different patterns of acute care access and outcomes are referred to. I expected that the third sentence would provide that explanation; however, the third sentence, starting with “More importantly” sounds like that it has moved to a different subject matter.

3. Again in the Discussion section, the second part of the following sentence is not clear: “In fact, our analysis may underestimate the degree of increased SUD associated harms as EMS data suggests increased out-of-hospital overdose-related cardiac arrests, which may in turn have not resulted in ED visits among people fearful of seeking emergency or hospital care (7).” Do you mean that the overdosed people with cardiac arrest refused transportation to ED or hospital care because they were afraid of contracting COVID?

4. “…increased trends of ED boarding (27)”: ED boarding may be an acceptable term among ED physicians, but it is not among non-ED readers.

5. Geographic differences: You cited the CDC’s provisional data on overdose death counts (Ahmad et al., 2021). The data show that many states in the West (CA, AZ, WA…) had higher increase in opioid and other drug overdose deaths in 2020. However, your Figure 3 data shows that the West region has the most muted increase of opioid and other drug related ED visits of all four regions. Does the difference suggest that fewer overdosed cases in the West were cared for at ED (i.e., EMT found the cases dead or DOA)? Or, is the difference attributable to that fact that your source of data are not representative of all EDs? Please refer to the following and provide some more discussion:

Baumgartner, J.C., and Radley, D.C., 2021. The spike in drug overdose deaths during the COVID-19 pandemic and policy options to move forward. https://www.commonwealthfund.org/blog/2021/spike-drug-overdose-deaths-during-covid-19-pandemic-and-policy-options-move-forward.

6. One last question: ED visit counts not number of unique individuals, right?

Reviewer #2: The authors analyzed a large sample of emergency department (ED) visits identified in the Clinical Emergency Data Registry (from 258 sites across 34 states) spanning 2019-2020. Key findings from the authors’ analyses were that overall ED visits declined in the early pandemic period and had not returned to baseline by the end of 2020; however, ED visits for substance use disorder (SUD) declined at slower rates in March and April 2020, such that the proportion of overall ED visits that were for SUD increased in the earlier months of 2020. The authors also found that ED visits for SUD returned to baseline in the second half of 2020. The authors concluded their paper with a meaningful discussion about hospital-based SUD care in 2020 within the context of resource allocation and availability concerns experienced by many hospital systems during 2020. Overall, the trends presented in this paper are meaningful within the context of public health surveillance and continuing to learn how the COVID-19 pandemic may have disrupted substance use disorder treatment.

However, there are several concerns and questions that should be addressed. I offer the following comments and questions for the authors’ consideration. I will first summarize key issues I have with the paper as currently written. Additional minor comments, suggestions, and questions follow.

Thank you for the opportunity to review your paper.

COMMENTS AND RECOMMENDATIONS

KEY ISSUES

A. General

1. My biggest concern is that other studies published prior to the submission of this paper have conducted similar analyses and highlighted some similar results. For example, Pines et al. (2021) examined ED visit trends for SUD from Jan-July 2020 and also found that ED visits for SUD decreased sharply early on, declined less than overall ED visits, and eventually began to return to 2019 levels. Pines et al. also explored differences by patient demographic factors. In turn, Holland et al. (2021) examined over 3,000 EDs spanning 48 states to explore trends in drug-related overdose/OUD and MH ED visit rates from Dec. 2018 to October 2020. Holland et al. described increases in OD-related ED visit rates, similarly noting that visits did not decrease in a manner similar to other ED visits, suggesting an increase in OUD ED burden in 2020.

I am not certain about the journal or editors’ stance on the novelty of findings or studies that coincide with or replicate other studies. However, it’s an important part of the scientific process, and surely the pandemic warrants much inquiry and better continued learning about how COVID-19 affects care delivery in various settings. Perhaps one thing the authors could do is cite studies like Holland et al., Pines et al., and Lucero et al. and differentiate in their paper how their analytic sample differs from others, if there are meaningful differences (e.g., significant differences in community-based EDs identified in ACEP CEDR?), and then reposition the Discussion to focus on the experience of those types of EDs, patient populations, etc., while also making sure to correctly summarize the gaps in the scholarly literature that are being filled.

References:

Pines, J. M., Zocchi, M. S., Black, B. S., Carlson, J. N., Celedon, P., Moghtaderi, A., & Venkat, A. (2021). How emergency department visits for substance use disorders have evolved during the early COVID-19 pandemic. Journal of Substance Abuse Treatment, 108391.

Holland, K. M., Jones, C., Vivolo-Kantor, A. M., Idaikkadar, N., Zwald, M., Hoots, B., ... & Houry, D. (2021). Trends in US emergency department visits for mental health, overdose, and violence outcomes before and during the COVID-19 pandemic. JAMA Psychiatry, 78(4), 372-379.

Lucero, A. D., Lee, A., Hyun, J., Lee, C., Kahwaji, C., Miller, G., ... & Pan, L. (2020). Underutilization of the emergency department during the COVID-19 pandemic. Western Journal of Emergency Medicine, 21(6), 15.

2. The authors seem to suggest their data will not be made publicly available. PLOS journals require authors to make all data necessary to replicate their study’s findings publicly available without restriction at the time of publication. Authors do not need to submit their entire data set if only a portion of the data was used in the reported study. When specific legal or ethical restrictions prohibit public sharing of a data set, authors must indicate how others may obtain access to the data. For studies involving third-party data, PLOS encourage authors to share any data specific to their analyses that they can legally distribute. PLOS recognizes, however, that authors may be using third-party data they do not have the rights to share. When third-party data cannot be publicly shared, authors must provide all information necessary for interested researchers to apply to gain access to the data. I.e., https://journals.plos.org/plosone/s/data-availability.

B. Introduction Section

1. As described in a later comment, I think the Introduction section nicely motivates the need for this type of analysis. However, it would be helpful if the authors directly stated the study objective in the second paragraph of the Introduction section. I understand you are just examining (important) trends and that the research question/study objective (and possibly any hypotheses) may be implicit. Although it is not necessary to test any explicit hypotheses in this type of study, given the journal’s broad scope and readership it would be helpful for the reader if you directly stated the study objective in relation to the gap you are ostensibly filling. E.g., perhaps open the second paragraph with an explicit statement of the objective, which would lead nicely into the rest of the paragraph supporting the rationale for that objective.

C. Methods Section

1. Did you use all ICD-10 codes within F10-F19 as inclusion criteria for SUD visits? Did you also include other relevant codes outside of that range, e.g., T40.-, G62.1, etc.? As supplemental content, could you include a table of all ICD-10 codes used to flag SUD and comparison conditions in this study, similar to Holland et al. (2021) and Pines et al. (2021)? This will be helpful for assessing/knowing what visits were included in the trend analyses, comparing to other studies, etc.

2. If I am interpreting this correctly, you summarized and described your analytic sample in sentences 2-4 of your Study Setting and Dataset (Methods) subsection. This is important information, but its placement in the Methods section is uncommon in health services research. These descriptive results would be better situated at the beginning of the Results section, helping the reader understand and assess your analytic sample characteristics leading into the main analyses (e.g., assessing generalizability and applicability of your analytic sample vis-a-vis achieving your study objective).

3. More information is needed about the regression modeling you performed. Just to confirm, the piecewise regression estimation process was simply used to identify the breakpoints, then you report the ED utilization in those identified periods? Could you provide a bit more information about the Poisson regression models used to estimate the IRRs comparing the 2020 periods to the 2019 reference periods? What variables went into the Poisson regression models? These were estimated for each period identified by the piecewise regression approach? At a minimum, more details are needed to assess what was conducted, though you could certainly include statistical notes as supplemental content to help the reader understand and replicate your analytic approach.

4. Tables and Figures: It would be helpful to include notes sections for each table and figure to briefly describe things like what the outcomes are/derived from which analytic method, number of observations, etc.

MINOR ISSUES AND COMMENTS

1. Please paginate your manuscript file. It makes providing feedback easier, given the files compiled and presented to reviewers.

2. Introduction: The beginning of the Introduction section is effective. The authors effectively summarized relevant (preliminary) findings from the scholarly literature, CDC, etc. The authors also quickly pivoted to motivating their research objective in a convincing way, noting the convergence of several critical public health crises (i.e., the “epidemic within a pandemic”).

3. Methods, Study Design subsection: I would recommend saying you conducted an “observational study” instead of “observational analysis”, then describe your analytic methods in your Analysis section.

4. Methods, Study Setting & Data subsection: PLOS encourages authors to cite any publicly available research data in their reference list, such as the U of Maryland COVID-19 Impact Analysis Platform. Please see: https://journals.plos.org/plosone/s/data-availability.

5. Results section: This section is well written, and key findings are discussed. However, the section is difficult to follow in relation to the tables and figures. I think you have indicated where you would like the placement of each table/figure. But I would recommend directly saying, “Table 1 shows….” or parenthetically citing the table/figure at the end of each sentence talking about the table/figure. E.g., “While visit rate for AUD… for the remainder of the year (Figure 2).

6. Do the x-axes on Figures 1 and S2 align? In Figure 1 it appears the 3-week moving average of ED visits overall begins to increase at the end 2020; however, that is not shown in Figure S2 (just what appears to be a new trend of decreasing moving averages). If different, does the increase (weeks 51-52?) affect the piecewise regression breakpoints?

7. Results section: You say, “ED visits for SUD and mental health exhibited similar, however, more muted declines in ED visitation and were much closer to baseline than overall visits by in the second half of 2020.” However, my read of Figure 1 is that SUD and MH ED 3-week moving averages also experienced precipitous decreases vis-à-vis 2019. If I am interpreting that correctly, I am not sure “muted” is the right word or tone for the discussion of this finding. Perhaps consider including the exact percentages (per 2019 visits) for SUD and MH ED visits as you did for all ED visits in the previous sentence. That will help clarify the relative rate of decrease.

8. You say, “Compared to similar periods in 2019, ED visits for SUD and AUD were close to baseline in the final period (Nov. 11 to Dec. 31) of 2020 (IRR 0.88 [0.85-0.91] and IRR 0.87 [0.83-0.91], respectively), while visits for OUD were above baseline (IRR 1.07 [1.02-1.13]).” I see how that can be interpreted from the IRR estimates presented in Table 1. However, the error bars in Figure 2 suggest the 3-week moving averages for OUD ED visits vis-à-vis the 2019 averages were not greater (i.e., crossing over 100% baseline). Am I interpreting that correctly? Or were the estimates greater than baseline for the average of that entire final breakpoint period in 2020? This is where the table/figure notes sections and additional statistical modeling information could be useful.

9. Figure 3 is interesting, though it is noisy and a bit difficult to interpret with the error bars. It is certainly responsible to illustrate the ranges, but I wonder if it might make sense just to plot the means for better conveying the key takeaways.

10. Discussion section: I think you meant to say “substantially higher rate” in the second sentence, not “substantial”.

11. Table 1 is nice. Very intuitive presentation.

12. The Discussion of potential explanations for your trends findings was compelling.

6. PLOS authors have the option to publish the peer review history of their article (what does this mean?). If published, this will include your full peer review and any attached files.

Reviewer #1: No

Reviewer #2: No

---

## [Author Response · Author response to Decision Letter 0]

4 Nov 2021

Dear Dr. Venkatesh,

Thank you for submitting your manuscript to PLOS ONE. After careful consideration, we feel that it has merit but does not fully meet PLOS ONE’s publication criteria as it currently stands. Therefore, we invite you to submit a revised version of the manuscript that addresses the points raised during the review process.

Please address the following journal requirements and additional editor and reviewer comments.

We look forward to receiving your revised manuscript.

Kind regards,

Fernando A. Wilson, PhD

Academic Editor

PLOS ONE

 

• We have updated the manuscript to meet the PLOS ONE style requirements

2. In ethics statement in the manuscript and in the online submission form, please provide additional information about the patient records/samples used in your retrospective study. Specifically, please ensure that you have discussed whether all data/samples were fully anonymized before you accessed them and/or whether the IRB or ethics committee waived the requirement for informed consent. If patients provided informed written consent to have data/samples from their medical records used in research, please include this information.

• We have added this to the end of the Methods section to clarify: “Analyses were performed in R (4.0.2). Access to the CEDR data is restricted given potentially identifying features, and the study was classified as exempt by the Institutional Review Board at Yale University.”

“This work was supported by the HHS Office of the Secretary Patient Centered Outcomes Research Trust Fund (PCORTF) under IDDA# ASPE-2018-001 and NIDA UG1DA015831-18S2. In addition, Dr. Venkatesh was supported by KL2 TR000140 from the National Center for Advancing Translational Sciences of the NIH and the American Board of Emergency Medicine – National Academy of Medicine Anniversary Fellowship. The contents of this work are solely the responsibility of the authors and do not necessarily represent the official view of NIH.”

• We have removed this from the acknowledgements in the revision per editorial guidance.

 “This work was supported by the HHS Office of the Secretary Patient Centered Outcomes Research Trust Fund (PCORTF) under IDDA# ASPE-2018-001 and NIDA UG1DA015831-18S2. In addition, AKV was supported by KL2 TR000140 from the National Center for Advancing Translational Sciences of the NIH and the American Board of Emergency Medicine – National Academy of Medicine Anniversary Fellowship. The contents of this work are solely the responsibility of the authors and do not necessarily represent the official view of NIH. The funders had no role in study design, data collection and analysis, decision to publish, or preparation of the manuscript.”

• We have removed funding-related text and updated the funding information in the submission portal.

• In the revised work, three separate data sources were included in this analysis. The University of Maryland COVID-19 Impact Analysis Platform, with county-level data on daily COVID-19 cases, is available on email request at this URL: https://data.covid.umd.edu/. The American Hospital Association Annual Survey data is available for purchase with data use agreement. However, the Clinical Emergency Data Registry (CEDR) dataset utilized is not a publicly available dataset, was obtained under a legal data use agreement to protect the identification of any site, physician group or physician and includes numerous easily identifiable elements were it available in a publicly accessible format. The dataset was not created or built upon any public funding or public disclosure requirements as well. Therefore, in accordance with the PLOS guidelines, this dataset cannot be available via public repository. 

a. If there are ethical or legal restrictions on sharing a de-identified data set, please explain them in detail (e.g., data contain potentially sensitive information, data are owned by a third-party organization, etc.) and who has imposed them (e.g., an ethics committee). Please also provide contact information for a data access committee, ethics committee, or other institutional body to which data requests may be sent.

• While the data fields for county-level COVID-19 cases are available via the University of Maryland COVID-19 Impact Analysis Platform, and data from the American Hospital Association Annual Survey are available for purchase, the Clinical Emergency Data Registry (CEDR) dataset is owned by a third party, the American College of Emergency Physicians (ACEP). Our data use agreement with ACEP permits the use of data for select research purposes under which no ED site, physician or physician group could be identified, which is not possible given the ease of reverse engineering identification if data were shared freely. Interested parties may seek to utilize the same dataset with permission by contacting the American College of Emergency Physicians at URL: https://www.acep.org/cedr.

Additional Editor Comments (if provided):

- The manuscript is not formatted according to PLOS ONE requirements, eg, include page and continuous line numbers. Refer to the following link:

https://journals.plos.org/plosone/s/submission-guidelines

- References are also not formatted following PLOS ONE requirements. Please review each reference and verify their formatting with the following instructions. Use square brackets for citations. Use NCBI journal abbreviations. For #2, please update this reference since it’s now published.

https://journals.plos.org/plosone/s/submission-guidelines#loc-references

https://www.ncbi.nlm.nih.gov/nlmcatalog/journals

• We have corrected the formatting of the citations in this resubmission.

- Add a dash within “co-occurrence”.

- Use either “U.S.” or “US” consistently throughout the manuscript.

• We have made further stylistic corrections as well.

- Use “1,000” instead of “1000” to be consistent with formatting of other numbers.

- Please provide more details on the regression analyses and regression tables in the Supporting Information section.

• We have added language to the Analysis section to specify that the segmented linear regression approach for overall ED visit counts ‘with time as the only input into the model’ was generated ‘using a previously described iterative linearization process for finding unknown breakpoints.’ Incident rate ratios (IRRs), reported to describe difference in incidence of ED visits across conditions in different time periods of analysis, were ‘obtained via unadjusted Poisson regression models.’

 

Reviewers' comments:

Reviewer's Responses to Questions

Comments to the Author

1. Is the manuscript technically sound, and do the data support the conclusions?

Reviewer #1: Yes

Reviewer #2: Partly

2. Has the statistical analysis been performed appropriately and rigorously? 

Reviewer #1: Yes

Reviewer #2: I Don't Know

3. Have the authors made all data underlying the findings in their manuscript fully available?

Reviewer #1: Yes

Reviewer #2: No

4. Is the manuscript presented in an intelligible fashion and written in standard English?

Reviewer #1: Yes

Reviewer #2: Yes

 

5. Review Comments to the Author

Reviewer #1: In this revised paper, the authors were very responsive to two previous reviewers’ comments and suggestions and addressed their concerns. The revised paper is solid in its analyses and presents interesting findings. Figures of changes overtime (compared to the 2019 baselines) are helpful for readers. However, writing in some parts, especially in the Discussion section, is difficult to understand (as previous reviewers also pointed out) and confusing. I also have a question about the findings related to geographic differences.

• We appreciate these sentiments and have sought to improve the clarity of this manuscript in this revision. 

1. The major source of confusion is that ED visit counts and rates are described in the same paragraph and it is not clear which one is which. I gathered that the % in the Y axis in the Figures is the proportion of ED visit counts in 2020 as compared to the same weeks in 2019. In the Results section (first paragraph—please specify page numbers), it is stated that “ED visits for SUD and mental health exhibited similar, however, more muted declines in ED visitation and were much closer to baseline than overall visits by in the second half of 2020.” And then, “While visit rates for AUD remained below 2019, ED utilization for OUD returned to 2019 levels by June, and were subsequently above 2019 baseline for the remainder of the year.” The “visit rate” is confusing, as it is not clear what you mean by that. Also clarify that the “baselines” are the ED visit counts in the corresponding weeks in 2019. Also, in “ED visits for SUD and mental health exhibited similar, however, more muted declines in ED visitation and were much closer to baseline than overall visits by in the second half of 2020,” you do not have to repeat “in ED visitation.”

• To improve clarity in this revision, we have removed the term “rate” and instead used language to describe the nominal/count of ED visits or described visits as “relatively” higher or lower when comparing to 2020 and 2019.

2. First paragraph of the Discussion section: “…ED visits for OUD have been increasing [at is missing here] a substantial higher rate than AUD or other SUDs since the beginning of the COVID-19 pandemic. These distinct condition specific trends demonstrate the fundamentally different patterns of acute care access and outcomes for different clinical scenarios. Most importantly, it is likely that a shift of focus away from the opioid epidemic as systems sought to manage and mitigate the COVID-19 pandemic expanded gaps in OUD treatment and naloxone access, driving increases in opioid-associated fatalities and exacerbating inequities in access to care.” I found last two sentences very confusing to read, as the second sentence is not specific about which the fundamentally different patterns of acute care access and outcomes are referred to. I expected that the third sentence would provide that explanation; however, the third sentence, starting with “More importantly” sounds like that it has moved to a different subject matter.

• We have corrected the typo mentioned above.

• We have attempted to clarify the intended point here, adding that:

“ED visits for OUD have been increasing at a substantially higher rate than AUD or other SUDs since the beginning of the COVID-19 pandemic. This suggests that the pandemic may have disproportionate condition specific impacts as a result of fundamentally different patterns of acute care access and outcomes for different clinical scenarios. For example, while the pandemic may have resulted in impeded access to outpatient treatment programs (OTPs) or office-based buprenorphine prescriptions that in turn resulted in poor OUD management and increased ED visitation, the pandemic may not have had as much relative effect on services for AUD that are less medication or healthcare facility dependent. It is likely that a shift of focus away from the opioid epidemic as systems sought to manage and mitigate the COVID-19 pandemic may have also expanded supply chain gaps, worsened contacts with healthcare providers necessary to ensure access to naloxone, potentially driving increases in opioid-associated fatalities and exacerbating inequities in access to care.”

3. Again in the Discussion section, the second part of the following sentence is not clear: “In fact, our analysis may underestimate the degree of increased SUD associated harms as EMS data suggests both increased transport re increased out-of-hospital overdose-related cardiac arrests, which may in turn have not resulted in ED visits among people fearful of seeking emergency or hospital care (7).” Do you mean that the overdosed people with cardiac arrest refused transportation to ED or hospital care because they were afraid of contracting COVID?

• We have added an additional reference regarding specifically deferred EMS transportation, and updated this to read and clarify: 

“In fact, our analysis may underestimate the degree of increased SUD associated harms as EMS data during the pandemic suggest both increased transport refusals for a myriad of conditions as well as increased out-of-hospital overdose-related cardiac arrests, which may not be classified as opioid related and may have resulted in transport to the ED previous to the pandemic.”

4. “…increased trends of ED boarding (27)”: ED boarding may be an acceptable term among ED physicians, but it is not among non-ED readers.

• We have replaced this phrase to read “increased trends of extremely prolonged ED length of stay, particularly the “boarding” of ED patients in the ED awaiting inpatient bed assignment” and further edited the language to improve clarity.

5. Geographic differences: You cited the CDC’s provisional data on overdose death counts (Ahmad et al., 2021). The data show that many states in the West (CA, AZ, WA…) had higher increase in opioid and other drug overdose deaths in 2020. However, your Figure 3 data shows that the West region has the most muted increase of opioid and other drug related ED visits of all four regions. Does the difference suggest that fewer overdosed cases in the West were cared for at ED (i.e., EMT found the cases dead or DOA)? Or, is the difference attributable to that fact that your source of data are not representative of all EDs? Please refer to the following and provide some more discussion:

Baumgartner, J.C., and Radley, D.C., 2021. The spike in drug overdose deaths during the COVID-19 pandemic and policy options to move forward. https://www.commonwealthfund.org/blog/2021/spike-drug-overdose-deaths-during-covid-19-pandemic-and-policy-options-move-forward.

• This is an excellent point and an important part of contextualizing our results. First, we acknowledge that more muted increase in OUD and other substance use disorder in the West may represent a limitation of the sample. That said, California is the best-represented site in the analysis with 41 hospital-based EDs. Second, it may be that the disconnect between ED visitation for OUD and population-level mortality represents a failure to activate emergency medical services, with consequences in the immediate phase and for patients who might have benefited from the initiation of medication-assisted therapy during an ED encounter for OUD:

“Interestingly, amongst the four US census regions reported in Fig 3, the West exhibited the smallest increases in ED presentations for OUD in our analysis. While this may represent a limitation of our work and ED visitation in our sample may not be representative generally, California was the best represented state in our sample (with 41 hospital-based EDs). This disconnect between ED visitation and population-level mortality may be related failure to active or delayed activation of EMS services. Given the value of EDs as sites for the initiation of medication-assisted therapy for OUD, failure to initiate transport in cases of opioid or other overdose not only risks mortality in the acute phase but also risks failure to initiate treatment and avoid a subsequent overdose.”

6. One last question: ED visit counts not number of unique individuals, right?

• Correct, our analysis is of ED visits not patients as some patients may have repeated ED visits. However as our analysis seeks to describe trends and does not include any outcome at risk of bias from patient-level clustering, a visit level analysis was felt to be most appropriate. 

Reviewer #2: The authors analyzed a large sample of emergency department (ED) visits identified in the Clinical Emergency Data Registry (from 258 sites across 34 states) spanning 2019-2020. Key findings from the authors’ analyses were that overall ED visits declined in the early pandemic period and had not returned to baseline by the end of 2020; however, ED visits for substance use disorder (SUD) declined at slower rates in March and April 2020, such that the proportion of overall ED visits that were for SUD increased in the earlier months of 2020. The authors also found that ED visits for SUD returned to baseline in the second half of 2020. The authors concluded their paper with a meaningful discussion about hospital-based SUD care in 2020 within the context of resource allocation and availability concerns experienced by many hospital systems during 2020. Overall, the trends presented in this paper are meaningful within the context of public health surveillance and continuing to learn how the COVID-19 pandemic may have disrupted substance use disorder treatment.

However, there are several concerns and questions that should be addressed. I offer the following comments and questions for the authors’ consideration. I will first summarize key issues I have with the paper as currently written. Additional minor comments, suggestions, and questions follow.

Thank you for the opportunity to review your paper.

COMMENTS AND RECOMMENDATIONS

KEY ISSUES

A. General

1. My biggest concern is that other studies published prior to the submission of this paper have conducted similar analyses and highlighted some similar results. For example, Pines et al. (2021) examined ED visit trends for SUD from Jan-July 2020 and also found that ED visits for SUD decreased sharply early on, declined less than overall ED visits, and eventually began to return to 2019 levels. Pines et al. also explored differences by patient demographic factors. In turn, Holland et al. (2021) examined over 3,000 EDs spanning 48 states to explore trends in drug-related overdose/OUD and MH ED visit rates from Dec. 2018 to October 2020. Holland et al. described increases in OD-related ED visit rates, similarly noting that visits did not decrease in a manner similar to other ED visits, suggesting an increase in OUD ED burden in 2020.

I am not certain about the journal or editors’ stance on the novelty of findings or studies that coincide with or replicate other studies. However, it’s an important part of the scientific process, and surely the pandemic warrants much inquiry and better continued learning about how COVID-19 affects care delivery in various settings. Perhaps one thing the authors could do is cite studies like Holland et al., Pines et al., and Lucero et al. and differentiate in their paper how their analytic sample differs from others, if there are meaningful differences (e.g., significant differences in community-based EDs identified in ACEP CEDR?), and then reposition the Discussion to focus on the experience of those types of EDs, patient populations, etc., while also making sure to correctly summarize the gaps in the scholarly literature that are being filled.

• We are sensitive to this concern and believe our work extends upon the prior cited work in several respects. First the Pines paper cited only captured data through July 2020, the Holland Paper through October 2020 and the Lucero paper through April 2020, which currently reflect only an incremental aspect of the initial pandemic. In this revision we include data through June 2021 demonstrating sustained changes throughout a dynamic pandemic, as well as heterogeneity in changes in visitation across OUD, alcohol-related visits, mental health visits, as compared to overall visits and visits for MI and stroke.

• Our paper also includes a more focused discussion and analysis of individual substance use disorders than some of the prior papers and includes a more heterogeneous mix of EDs than the paper of Pines and Lucero, both of which are based on the data of single contract management groups in EM. Therefore our findings may be more generalizable.

• We have updated the revision to not only include these citations but also discuss the importance of our study in extending these findings through June 30, 2021.

References:

Pines, J. M., Zocchi, M. S., Black, B. S., Carlson, J. N., Celedon, P., Moghtaderi, A., & Venkat, A. (2021). How emergency department visits for substance use disorders have evolved during the early COVID-19 pandemic. Journal of Substance Abuse Treatment, 108391.

Holland, K. M., Jones, C., Vivolo-Kantor, A. M., Idaikkadar, N., Zwald, M., Hoots, B., ... & Houry, D. (2021). Trends in US emergency department visits for mental health, overdose, and violence outcomes before and during the COVID-19 pandemic. JAMA Psychiatry, 78(4), 372-379.

Lucero, A. D., Lee, A., Hyun, J., Lee, C., Kahwaji, C., Miller, G., ... & Pan, L. (2020). Underutilization of the emergency department during the COVID-19 pandemic. Western Journal of Emergency Medicine, 21(6), 15.

2. The authors seem to suggest their data will not be made publicly available. PLOS journals require authors to make all data necessary to replicate their study’s findings publicly available without restriction at the time of publication. Authors do not need to submit their entire data set if only a portion of the data was used in the reported study. When specific legal or ethical restrictions prohibit public sharing of a data set, authors must indicate how others may obtain access to the data. For studies involving third-party data, PLOS encourage authors to share any data specific to their analyses that they can legally distribute. PLOS recognizes, however, that authors may be using third-party data they do not have the rights to share. When third-party data cannot be publicly shared, authors must provide all information necessary for interested researchers to apply to gain access to the data. I.e., https://journals.plos.org/plosone/s/data-availability.

• We addressed this above.

B. Introduction Section

1. As described in a later comment, I think the Introduction section nicely motivates the need for this type of analysis. However, it would be helpful if the authors directly stated the study objective in the second paragraph of the Introduction section. I understand you are just examining (important) trends and that the research question/study objective (and possibly any hypotheses) may be implicit. Although it is not necessary to test any explicit hypotheses in this type of study, given the journal’s broad scope and readership it would be helpful for the reader if you directly stated the study objective in relation to the gap you are ostensibly filling. E.g., perhaps open the second paragraph with an explicit statement of the objective, which would lead nicely into the rest of the paragraph supporting the rationale for that objective.

• We have made changes to the final paragraph of the Introduction to better frame the objective of the study:

“Given the gradual emergence of a ‘new normal’ and ongoing concerns about excess mortality both directly attributable to COVID-19 and related to SUDs, we aimed to provide an updated characterization of ED visitation for SUD in a national sample of hospital-based EDs and differentiate from prior work with a more granular exploration specifically identifying OUD, mental health visits, and comparing these with overall visits and visits for select emergent medical conditions.”

C. Methods Section

1. Did you use all ICD-10 codes within F10-F19 as inclusion criteria for SUD visits? Did you also include other relevant codes outside of that range, e.g., T40.-, G62.1, etc.? As supplemental content, could you include a table of all ICD-10 codes used to flag SUD and comparison conditions in this study, similar to Holland et al. (2021) and Pines et al. (2021)? This will be helpful for assessing/knowing what visits were included in the trend analyses, comparing to other studies, etc.

• In response to this thoughtful suggestion we have actually revised our methods to follow those of Pines et al. (2021) by making use of the Clinical Classification Software Revised codes for substance use as in their work now explained in the Methods section. This did add a small number of codes and visits to the sample, without a major effect on our overall findings

• We have included a supplemental table as requested by the reviewers for the diagnostic codes used to identify each group of visits in the analysis.

3. If I am interpreting this correctly, you summarized and described your analytic sample in sentences 2-4 of your Study Setting and Dataset (Methods) subsection. This is important information, but its placement in the Methods section is uncommon in health services research. These descriptive results would be better situated at the beginning of the Results section, helping the reader understand and assess your analytic sample characteristics leading into the main analyses (e.g., assessing generalizability and applicability of your analytic sample vis-a-vis achieving your study objective).

• We have moved these analytic sample descriptions to the Results Section as requested.

4. More information is needed about the regression modeling you performed. Just to confirm, the piecewise regression estimation process was simply used to identify the breakpoints, then you report the ED utilization in those identified periods? Could you provide a bit more information about the Poisson regression models used to estimate the IRRs comparing the 2020 periods to the 2019 reference periods? What variables went into the Poisson regression models? These were estimated for each period identified by the piecewise regression approach? At a minimum, more details are needed to assess what was conducted, though you could certainly include statistical notes as supplemental content to help the reader understand and replicate your analytic approach.

• We have added additional detail to the methods regarding the regression approach and clarifying that the piecewise regression estimation process was simply used to identify breakpoints to report ED utilization in those identified periods. The Poisson models were unadjusted, estimated at the level of the hospital-based EDs in the sample simply for the purpose of reporting incident rate ratios for visit counts comparing different time periods in 2020/2021 to 2019.

5. Tables and Figures: It would be helpful to include notes sections for each table and figure to briefly describe things like what the outcomes are/derived from which analytic method, number of observations, etc.

• We have updated this draft to includes footnotes for the Figures and Table to better clarify the results.

MINOR ISSUES AND COMMENTS � 

1. Please paginate your manuscript file. It makes providing feedback easier, given the files compiled and presented to reviewers.

2. Introduction: The beginning of the Introduction section is effective. The authors effectively summarized relevant (preliminary) findings from the scholarly literature, CDC, etc. The authors also quickly pivoted to motivating their research objective in a convincing way, noting the convergence of several critical public health crises (i.e., the “epidemic within a pandemic”).

• We appreciate the positive feedback.

3. Methods, Study Design subsection: I would recommend saying you conducted an “observational study” instead of “observational analysis”, then describe your analytic methods in your Analysis section.

• Agreed, language adjusted as suggested.

4. Methods, Study Setting & Data subsection: PLOS encourages authors to cite any publicly available research data in their reference list, such as the U of Maryland COVID-19 Impact Analysis Platform. Please see: https://journals.plos.org/plosone/s/data-availability.

• Agreed, the citation has been added.

5. Results section: This section is well written, and key findings are discussed. However, the section is difficult to follow in relation to the tables and figures. I think you have indicated where you would like the placement of each table/figure. But I would recommend directly saying, “Table 1 shows….” or parenthetically citing the table/figure at the end of each sentence talking about the table/figure. E.g., “While visit rate for AUD… for the remainder of the year (Figure 2).

• Agreed, the results section has been edited to reflect this suggestion.

6. Do the x-axes on Figures 1 and S2 align? In Figure 1 it appears the 3-week moving average of ED visits overall begins to increase at the end 2020; however, that is not shown in Figure S2 (just what appears to be a new trend of decreasing moving averages). If different, does the increase (weeks 51-52?) affect the piecewise regression breakpoints?

• The Figures have been adjusted so that x-axis align. There is a decline in the last few weeks of 2020 exhibited in both Figure 1 and Supplemental Figure 2.

7. Results section: You say, “ED visits for SUD and mental health exhibited similar, however, more muted declines in ED visitation and were much closer to baseline than overall visits by in the second half of 2020.” However, my read of Figure 1 is that SUD and MH ED 3-week moving averages also experienced precipitous decreases vis-à-vis 2019. If I am interpreting that correctly, I am not sure “muted” is the right word or tone for the discussion of this finding. Perhaps consider including the exact percentages (per 2019 visits) for SUD and MH ED visits as you did for all ED visits in the previous sentence. That will help clarify the relative rate of decrease.

• Agreed, the results section has been edited to reflect this suggestion.

8. You say, “Compared to similar periods in 2019, ED visits for SUD and AUD were close to baseline in the final period (Nov. 11 to Dec. 31) of 2020 (IRR 0.88 [0.85-0.91] and IRR 0.87 [0.83-0.91], respectively), while visits for OUD were above baseline (IRR 1.07 [1.02-1.13]).” I see how that can be interpreted from the IRR estimates presented in Table 1. However, the error bars in Figure 2 suggest the 3-week moving averages for OUD ED visits vis-à-vis the 2019 averages were not greater (i.e., crossing over 100% baseline). Am I interpreting that correctly? Or were the estimates greater than baseline for the average of that entire final breakpoint period in 2020? This is where the table/figure notes sections and additional statistical modeling information could be useful.

• This is an excellent point for clarification. In our updated manuscript, viewed in aggregate and estimated with a simple unadjusted Poisson regression model for the periods for weeks 28-53 of 2020 and weeks 1-12 of 2021, OUD visits were above baseline. These differences, when comparing individual 3-week moving averages, were generally not statistically significantly different from 2019 levels as depicted in Figure 2.

9. Figure 3 is interesting, though it is noisy and a bit difficult to interpret with the error bars. It is certainly responsible to illustrate the ranges, but I wonder if it might make sense just to plot the means for better conveying the key takeaways.

• Agreed, for the purpose of readability Figure 3 no longer includes the error bars.

10. Discussion section: I think you meant to say “substantially higher rate” in the second sentence, not “substantial”.

• Fixed.

11. Table 1 is nice. Very intuitive presentation.

• We appreciate the positive feedback.

12. The Discussion of potential explanations for your trends findings was compelling.

• We appreciate the positive feedback.

---

## [Editor Report · Decision Letter 1]

17 Dec 2021

Emergency Department Utilization for Substance Use Disorders and Mental Health Conditions During COVID-19

PONE-D-21-20023R1

Dear Dr. Venkatesh,

We’re pleased to inform you that your manuscript has been judged scientifically suitable for publication and will be formally accepted for publication once it meets all outstanding technical requirements.

Kind regards,

Fernando A. Wilson, PhD

Academic Editor

PLOS ONE

Additional Editor Comments (optional):

- At the next available opportunity during production, please ensure that Supplemental Table 1 is uploaded, and insert "of" between "myriad" and "conditions" in Discussion.
---

## [Editor Report · Acceptance letter]

5 Jan 2022

PONE-D-21-20023R1 

Emergency Department Utilization for Substance Use Disorders and Mental Health Conditions During COVID-19 

Dear Dr. Venkatesh:

I'm pleased to inform you that your manuscript has been deemed suitable for publication in PLOS ONE. Congratulations! Your manuscript is now with our production department. 

Kind regards, 

on behalf of

Dr. Fernando A. Wilson 

Academic Editor

PLOS ONE